# Reduced MIP-1β as a Trait Marker and Reduced IL-7 and IL-12 as State Markers of Anorexia Nervosa

**DOI:** 10.3390/jpm11080814

**Published:** 2021-08-20

**Authors:** Johanna Louise Keeler, Olivia Patsalos, Raymond Chung, Ulrike Schmidt, Gerome Breen, Janet Treasure, Hubertus Himmerich, Bethan Dalton

**Affiliations:** 1Section of Eating Disorders, Department of Psychological Medicine, Institute of Psychiatry, Psychology and Neuroscience, King’s College London, London SE5 8AF, UK; olivia.patsalos@kcl.ac.uk (O.P.); ulrike.schmidt@kcl.ac.uk (U.S.); janet.treasure@kcl.ac.uk (J.T.); hubertus.himmerich@kcl.ac.uk (H.H.); bethan.dalton@kcl.ac.uk (B.D.); 2MRC Social, Genetic and Developmental Psychiatry Centre, Institute of Psychiatry, Psychology and Neuroscience, King’s College London, London SE5 8AF, UK; raymond.chung@kcl.ac.uk (R.C.); gerome.breen@kcl.ac.uk (G.B.); 3South London and Maudsley NHS Foundation Trust, Bethlem Royal Hospital, Monks Orchard Road, Beckenham, Kent BR3 3BX, UK

**Keywords:** anorexia nervosa, cytokines, chemokines, brain-derived neurotrophic factor, cross-sectional, inflammatory markers

## Abstract

Alterations in certain inflammatory markers have been found in individuals with anorexia nervosa (AN). However, their relation to clinical characteristics has not been extensively explored, nor is it clear whether they are trait or state features of the disorder. This cross-sectional study measured serum concentrations of 36 inflammatory markers in people with acute AN (*n* = 56), recovered AN (rec-AN; *n* = 24) and healthy controls (HC; *n* = 51). The relationship between body mass index (BMI), eating disorder psychopathology, depression symptoms and inflammatory markers was assessed. Statistical models controlled for variables known to influence cytokine concentrations (i.e., age, ethnicity, smoking status and medication usage). Overall, most inflammatory markers including pro-inflammatory cytokines were unchanged in AN and rec-AN. However, in AN and rec-AN, concentrations of macrophage inflammatory protein (MIP)-1β were lower than HCs. Interleukin (IL)-7 and IL-12/IL-23p40 were reduced in AN, and concentrations of macrophage-derived chemokine, MIP-1α and tumor necrosis factor-α were reduced in rec-AN compared to HC. In conclusion, a reduction in MIP-1β may be a trait marker of the illness, whereas reductions in IL-7 and IL-12/IL-23p40 may be state markers. The absence of increased pro-inflammatory cytokines in AN is contradictory to the wider literature, although the inclusion of covariates may explain our differing findings.

## 1. Introduction

Anorexia nervosa (AN) is a serious and persistent psychiatric disorder characterised by a low body weight due to food restriction, body image disturbance and intense fear of weight gain [1]. The precise aetiology of AN is not clear, although biopsychosocial models acknowledge the interaction of genetic, societal and environmental factors, together with biological factors such as the immune system [2]. Altered concentrations of inflammatory markers, including cytokines, have previously been reported in AN [3]. Cytokines are signalling proteins that are produced by a range of immune cells in the periphery and brain (e.g., microglia and astrocytes) [4]. They play a crucial role in the regulation of the immune system, in the pathophysiology of autoimmune disorders and in brain development and function [5]. To date, there have been no clearly identified state or trait inflammatory markers and few studies have attempted to explore the link between inflammatory markers and clinical characteristics associated with the disorder.

Cytokines can be broadly categorised according to their immunological function, into TH1 cytokines (interferon (IFN)-gamma (IFN-γ), interleukin (IL)-2 and IL-12), TH2 cytokines (IL-4, IL-5 and IL-13), pro-inflammatory cytokines (IL-1, IL-6, IL-8, IL-17, IL-21, IL-22, IFN-α and tumor necrosis factor (TNF)-α), and anti-inflammatory cytokines (IL-10, transforming growth factor (TGF)-β) [6]. Additionally, chemokines are a family of small cytokines that coordinate immune cells to attract them to sites of inflammation. Pro-inflammatory cytokines, which are involved in upregulating the inflammatory response, have been the focus of research in AN.

Meta-analyses of in vivo studies have found evidence for increases in certain pro-inflammatory cytokines, such as TNF-α, IL-6 and IL-1β, in AN [3,7]. Certain indicators of illness severity, such as BMI, are known to affect cytokine concentrations, with extremely high or low BMI resulting in high levels of pro-inflammatory cytokines [7,8]. There is evidence that alterations in cytokine concentrations are partially reversible with weight gain; for example, levels of IL-6 and IL-7 have been shown to normalise after weight gain above a body mass index (BMI) of 18.5 kg/m^2^ [7,9]. In a longitudinal study, this decrease in IL-6 concentration co-occurred with small improvements in psychological eating disorder symptoms [9], indicating a potential state biomarker of AN. However, despite improvements in weight, certain cytokines often remain altered in these individuals (e.g., TNF-α and IL-1β; [7]), indicative of possible trait markers of the illness. It is unclear whether other cytokines implicated in the acute stages of AN (e.g., TNF-β and IL-15; [10]) are state or trait markers of AN. A recent cross-sectional study examined inflammatory markers in AN, recovered AN (rec-AN) patients and healthy controls (HCs) [11]. Results indicated differences in several inflammatory markers in AN compared to the controls (e.g., lower concentrations of components of TNF, IL-12β, IL-18 receptor-β, IL-10 receptor-β), but no differences between rec-AN and HCs. Many of the inflammatory markers found to be altered in the acute AN group were related to BMI. The authors concluded that the aberrant inflammatory profile seen in the acute stages of AN is a state marker and is normalised in those recovered from the disorder [11]. However, inflammatory markers were not explored in relation to clinical symptoms or other psychopathology (e.g., depression), which potentially could contribute to alterations in the inflammatory profile of the acute AN patient.

There are high rates of comorbidity in AN, with one study finding over two-thirds of patients having a comorbid Axis 1 disorder, such as major depressive disorder or an anxiety disorder [12]. Moreover, traumatic childhood experiences are a risk factor for the development of an eating disorder [13], and post-traumatic stress disorder (PTSD) is prevalent in approximately 15–25% of AN cases [14,15]. Increases in pro-inflammatory cytokines (e.g., TNF-α, IL-6, IL-1β) are thought to be involved in the pathogenesis of many psychiatric disorders, including depression, schizophrenia, addiction and PTSD [16,17]. Furthermore, antipsychotic and antidepressant medication, which is used to treat these comorbidities, has also been reported to lead to changes of cytokine production and signalling in vivo [18,19] and in vitro [20,21].

Investigating the role of psychiatric symptoms, particularly those that have been associated with altered cytokine concentrations (e.g., trauma, depression, stress and anxiety) [22], may elucidate the relationship between psychological variables and cytokine concentrations in AN. A previous study in our group assessed the relationship between clinical variables and cytokine concentrations in AN [10], identifying relationships between eating disorder psychopathology and inflammatory markers (e.g., IFN-γ–inducible protein 10 (IP-10), placental growth factor) and general psychopathology and other inflammatory markers (e.g., Eotaxin, IL-7, IL-8, IP-10, monocyte chemoattractant protein (MCP)-1, thymus- and activation-regulated chemokine (TARC)). However, this study was not extended to individuals recovered from AN, and it also identified key confounding variables (e.g., age and BMI), which were not controlled for in the analyses.

Additionally, the focus on pro-inflammatory cytokines in the literature has meant that other groups of cytokines have not been explored. For example, there has been little emphasis on cytokines expressed by T-helper type (Th17) cells, which include IL-17α, IL-21 and IL-22 [23]. These cytokines are implicated in autoimmune conditions and inflammatory processes, where their production can lead to excessive inflammation (IL-17α; [24]) and contribute to the pathogenesis of autoimmune diseases [25]. Autoimmune diseases are thought to have bidirectional links with eating disorders; a diagnosis of one condition increases the risk for a diagnosis of the other [26], although the mediating factors are unclear. IL-17 has also been linked to the presence of anxiety symptoms in patients with autoimmune disease [27], as well as the pathogenesis and maintenance of other psychiatric disorders [20,28]. Similarly, there has been little investigation of the role of chemokines in AN. Chemokines (e.g., monocyte chemoattractant protein (MCP)-1, macrophage inflammatory protein (MIP)-1α, MIP-1β, RANTES) are thought to have a generalised involvement across a range of psychiatric disorders [29] and have a neuromodulatory effect that can alter cognition [30]. Moreover, alterations in chemokine function have been linked to depression through their role in the regulation of adult hippocampal neurogenesis and neuroplasticity [31], which is thought to be altered in AN [32,33].

Taken together, whilst there is replicated evidence for alterations of specific cytokines in AN (e.g., IL-6, TNF-α, IL-1β), there are other cytokines and chemokines that have not been investigated as thoroughly in this population (e.g., IL-7, IL-12, IL-17α, IL-21, IL-22, MCP-1, MIP-1α, MIP-1β) or are lacking a confirmatory study. Furthermore, the extent to which inflammatory marker concentrations are related to characteristics associated with the clinical features of AN has not been fully investigated. Altered concentrations of inflammatory markers may not only be of scientific interest as biomarkers or pivotal messenger molecules involved in the pathophysiology of AN. They may also represent future drug targets, because inhibitors of certain cytokine pathways are available and approved for the treatment of autoimmune disorders, and they have also been shown to influence weight in meta-analytic research [34,35]. Additional treatments such as non-steroidal anti-inflammatory drugs, omega-3 fatty acids, statins and minocyclines have shown anti-inflammatory effects in major depressive disorder [36]. Thus, these drugs could be repurposed for use in AN.

In this exploratory study, using an array of 36 inflammatory markers, we aimed to (a) determine whether, and which, cytokines differ between individuals with AN, rec-AN, and HCs and (b) determine how the levels of inflammatory markers in individuals with AN are related to various clinical characteristics such as BMI, illness duration, eating disorder symptom severity and psychiatric comorbidities.

## 2. Materials and Methods

### 2.1. Participants and Study Design

A total of 142 females over the age of 18 with AN (*n* = 58) or rec-AN (*n* = 25), or healthy individuals with no current or previous diagnosis of a psychiatric disorder (*n* = 59), took part in this study. Participants with AN were recruited through Eating Disorder Services at the South London and Maudsley NHS Foundation Trust, online advertisements or participation in other studies at King’s College London (KCL). Those in the AN group were required to have a current diagnosis of AN according to the DSM-5 criteria [1] and a BMI of < 18.5 kg/m^2^. The rec-AN and HC groups were recruited through online and community advertisements and via their participation in other studies at KCL. The rec-AN group had to have previously met the criteria for a diagnosis of AN, based on the DSM-5 criteria [1], to have maintained a BMI above 18.5 kg/m^2^ and to not have engaged in any disordered eating behaviours for the past six months. HCs were required to have a BMI above 18.5 kg/m^2^ and have no history of or current psychiatric diagnosis. Exclusion criteria for all participants were as follows: male sex; under the age of 18; having any acute or chronic inflammatory or autoimmune condition; self-report of recent infection; current pregnancy; excessive alcohol consumption (>3 units/day, 5 days a week) and/or cigarette consumption (>15 cigarettes/day).

Participants underwent telephone screening. This included the Eating Disorder Diagnostic Scale (EDDS; [37]) to identify eating disorder psychopathology, a brief assessment of physical health conditions and an inclusion/exclusion screen specific to this study. Additionally, HCs completed the Structured Clinical Interview for DSM-IV Axis I Disorders [38] to assess for current or past psychiatric conditions.

Following screening, participants were invited to a study session at King’s College London where they completed questionnaires assessing demographic data (e.g., ethnicity, smoking status, current medication) and clinical data (e.g., illness duration, lowest weight, symptom-free periods). Participants also completed a battery of questionnaires measuring eating disorder and related psychopathology (see Section 2.2. for specific questionnaires used in the present study). Participants’ height and weight were measured, together with body fat via a non-invasive Inbody S10 machine, Biospace Co., Ltd. Finally, blood samples, including one 5 mL serum tube for use in the current study, were collected from participants by trained phlebotomists on behalf of the NIHR BRC BioResource.

An *a priori* power analysis was conducted to calculate the number of required participants using the means and standard deviations of a previous study [10]. The analysis was conducted using the following: a one-way analysis of covariance (ANCOVA) with independent samples, an effect size of 0.8, a power level of 0.8 and a level of significance of 0.00135, to account for Bonferroni corrections for multiple testing of 36 cytokine tests. This calculation yielded a total sample of 49 (17 per group). Therefore, our sample exceeds this minimum sample size calculation and should be sufficient to detect an effect.

### 2.2. Measures

#### 2.2.1. Questionnaires

Eating disorder (ED) symptoms over the past 28 days were assessed using the Eating Disorder Questionnaire Version 6.0 (EDE-Q 6.0; [39]). The Depression, Anxiety and Stress Scale—21 (DASS-21; [40]) assessed general psychopathology over the past week and the Beck Depression Inventory 2 (BDI-II; [41]) assessed depression symptoms over the previous two weeks. Symptoms of PTSD were assessed using the PTSD Checklist for DSM-5 (PCL5; [42]), with an accompanying questionnaire assessing for adverse life events: the Life Events Checklist for DSM-5 (LEC-5; [43]). The LEC-5 was used to verify exposure to a potential trauma. Aligned with the DSM-5 diagnostic criteria for PTSD, participants were coded as having been exposed to a traumatic event (direct experience, witnessing, learning that the traumatic event occurred to a close family member or close friend or repeated/extreme exposure to details of the traumatic event). A score of 32 or above was indicative as probable PTSD in response to this traumatic event [44].

#### 2.2.2. Cytokine Measurement

Blood samples were processed by experienced laboratory staff at the Social, Genetic and Developmental Psychiatry Centre (SGDP) at King’s College London (KCL). Serum was separated with centrifugation, stored at −80 °C prior to use and thawed at room temperature. Samples were anonymised and stored securely. Concentrations of 36 inflammatory markers were quantified simultaneously using multiplex ELISA-based technology, provided by the Meso Scale Discovery V-PLEX Plus Human Biomarker 36-Plex Kit, following manufacturer instructions (Meso Scale Discovery, Maryland, USA). Cytokines measured in the assay included the following: Eotaxin, Eotaxin-3, granulocyte-macrophage colony-stimulating factor (GM-CSF), IFN-γ, IL-1α, IL-1β, IL-2, IL-4, IL-5, IL-6, IL-7, IL-8, IL-8 (HA), IL-10, IL-12/IL-23p40, IL-12p70, IL-13, IL-15, IL-16, IL-17α, IL-21, IL-22, IL-23, IL-27, IL-31, IP-10, MCP-1, MCP-4, macrophage-derived chemokine (MDC), MIP-1α, MIP-1β, MIP- 3α, TARC, TNF-α, TNF-β and vascular endothelial growth factor (VEGF)-α. IL-12p70 is the active heterodimer of IL-12. IL-12 is a secreted heterodimeric cytokines that contains disulfide-linked p35 and p40 subunits, the latter of which is also a subunit of the heterodimeric IL-23.

Seven-point standard curves were run in duplicate on each plate to calculate the absolute pg/mL values of cytokines for each sample. Cases and controls were randomised across batches, which were scanned using the Meso Scale Discovery MESO Quickplex SQ 120 reader at the SGDP centre at KCL. Measurement of IL-17A and IL-8 were duplicated across two plates, and analyses were performed on the duplicate with the lowest proportion of samples with undetectable concentrations of cytokines.

### 2.3. Statistical Analysis

A total of nine participants were excluded from analyses due to blood samples not being collected. Moreover, two participants reported smoking more than 15 cigarettes a day, and so were excluded from analyses. Therefore, data from a total of 131 participants were included in all analyses (HC = 51; AN = 56; rec-AN = 24).

Standard curves were used to ascertain absolute values (pg/mL) of the cytokines. All statistical analyses were performed in SPSS [45]. For all analyses, effect sizes are reported with Cohen’s *d* using thresholds of 0.2 as indicative of a small effect, 0.5 as a moderate effect and 0.8 as a large effect. Results are discussed where effect sizes are of a moderate size or larger.

#### 2.3.1. Cross-Sectional Demographic, Clinical Characteristic and Cytokine Comparisons

Baseline differences in clinical and demographic characteristics were ascertained using analysis of variance (ANOVA) models, and Tukey post-hoc tests. Variables that violated a Levene’s Test of Equality of Error Variances were transformed by Log10 transformation before comparison. If transformation did not correct heterogeneity in variance, non-parametric tests were used (e.g., Kruskal–Wallis test). Post-hoc comparisons were Bonferroni corrected where appropriate. Differences in nominal variables (e.g., smoking status, drug use, ethnicity) were assessed using the chi-square test of homogeneity.

Several cytokines (*n* = 15) had missing values ranging from 6% to 92% (see Appendix A for a summary of missingness per cytokine). Missing data for cytokines with less than 50% non-detectable data were replaced with the minimum detection limit for each respective cytokine. Cytokines with between 50–90% non-detectable data (IL-1α, IL-1β, IL-5 and IL-21) were excluded from analyses, and chi-square tests were used to examine differences between groups in the distribution of detectable to non-detectable values. For IL-23, 92% of values were non-detectable, and so this cytokine was excluded from analyses. Upper and lower detection limits are shown in Appendix A.

Separate ANCOVA models were used to explore cytokine differences between groups, using predetermined covariates. Variables that influence cytokine concentrations include, age, ethnicity, medication usage and smoking status [46]. Cytokine concentrations tend to increase over the lifespan [47]. Moreover, variations in cytokines broadly occur across ethnic groups, with evidence for some ethnic groups having greater levels of pro-inflammatory cytokines [48]. For example, TNF-α levels were higher in Mexican Americans than white Americans [49], and African Americans showed higher levels of IL-6 than white Americans (e.g., [50]). Finally, cytokine levels may be modulated with the use of psychiatric medications that are often prescribed in the AN population, such as selective-serotonin reuptake inhibitors (SSRIs), which have been found to reduce levels of IL-6 and TNF-α [51]. Thus, age, smoking status (i.e., smoker/non-smoker), psychopharmacological medication use (i.e., present/absent) and ethnicity (i.e., white, ethnic minority) were used as covariates.

The Shapiro–Wilk test of normality was used to assess residuals for normality of distribution. If residuals were not normally distributed, they were transformed with log10 transformation and used in an ANCOVA model. If residuals were still non-normally distributed, data were assessed using the nonparametric analysis of covariance (Quade’s method). The Benjamini–Hochberg procedure was applied to correct for multiple comparisons, with a false discovery rate (FDR) of 0.1. Both raw cytokine values and adjusted marginal means based on transformed concentrations are presented.

#### 2.3.2. Exploratory Regression Analyses

Exploratory regression analyses were performed on the full sample, using BMI, EDE-Q Global Score and BDI-II Total score as predictors and inflammatory marker concentrations as the dependent variable, whilst controlling for age, ethnicity, use of psychopharmacological medications and smoking status. In the AN group, separate regression analyses were performed, using BMI, EDE-Q Global Score, BDI-II Total score, illness duration (time since diagnosis) and probable PTSD as individual predictors and inflammatory marker concentrations as the dependent variable, again controlling for age, ethnicity, smoking status and psychopharmacological medication use. Separate regression models were conducted for each clinical characteristic and inflammatory marker, with standardised coefficients, confidence intervals and *p*-values reported. Each inflammatory marker was log10 transformed to achieve approximately normal distribution. Studentised residuals greater than +3 standard deviations were deemed outliers and were removed, which was noted. The Benjamini–Hochberg procedure with an FDR of 0.1 was used per clinical characteristic to correct significance values for multiple testing.

## 3. Results

Demographic and clinical characteristics for the AN, rec-AN and HCs are presented in Table 1. Age significantly differed between groups, with the HC group being significantly younger than the AN group (*p* = 0.015; *d* = 0.43). Moreover, the groups were different in terms of ethnicity (*p* < 0.01; *d* = 1.11), whereby the HC group had a higher proportion of ethnic minorities. As expected, BMI and body fat percentage significantly differed between the groups, with the AN group being significantly lower than the rec-AN (both *p* < 0.001; *d* < 1.4) and HC groups (both *p* < 0.001; *d* < 2.0). Additionally, the proportion of participants with a diagnosis of restrictive versus binge-purge AN differed between the AN and rec-AN groups. In the AN group, 39 participants (70%) were in current psychological treatment; of which one (3%) was an inpatient and 28 (72%) were receiving outpatient care.

In terms of measures of psychopathology, EDE-Q Global scores were greater in AN than the rec-AN and HC groups (all *p* < 0.001; *d* < 2.0). Depression severity, as measured by the BDI-II, was greater in AN than rec-AN (*p* < 0.001; *d* = 1.53) and greater in rec-AN than HC (*p* = 0.001; *d* = 0.61). Total scores on the DASS were greater in AN than HCs (*p* < 0.001; *d* = 2.28) and were greater in rec-AN than HCs (*p* = 0.001; *d* = 0.63). Scores on the DASS were indicative of severe depression (which is in agreement with BDI-II scores), moderate stress and severe anxiety in the AN group.

### 3.1. Cross-Sectional Comparisons of Inflammatory Markers

Four cytokines (IL-1α, IL-1β, IL-5 and IL-21) showed greater than 50% missing data. Chi-square tests revealed no significant differences between groups in the proportion of undetectable values, for any of the cytokines. Therefore, 30 inflammatory markers were included in the analyses thereafter. Known quantities within the standard curves correlated highly with quantities predicted by fluorescence intensity (r2 > 0.99) for all measured cytokines. Results of the cross-sectional comparisons are displayed schematically in Figure 1A. Detailed results of the ANCOVAs can be seen in Table 2.

#### 3.1.1. Differences between the AN Groups and Healthy Controls

Figure 1 shows cross-sectional comparisons, using ANCOVA, in cytokine values between AN, rec-AN and HC groups, controlling for age, ethnicity, psychopharmacological medication usage and smoking status. The raw and adjusted means and standard deviations, together with *p*-values and effect sizes, are presented in Table 2. IL-7 and IL-12/IL-23p40 were lower in AN than HCs, with a moderate effect size. Moreover, MIP-1β was significantly lower in both the AN and rec-AN groups compared to HCs, with a moderate effect size.

The rec-AN group were also lower than HCs in Eotaxin-3, with a large effect size, and MDC, TNF-α and MIP-1α, with a moderate effect size.

#### 3.1.2. Differences between Acute and Recovered AN Groups

The rec-AN group showed lower concentrations of Eotaxin-3, with a large effect size, and IL-22, MCP-1, MDC and MIP-1α, with a moderate effect size, than the AN group. In contrast, concentrations of GM-CSF and IL-16 were moderately higher in the rec-AN compared to the AN group.

### 3.2. Exploratory Regressions between Inflammatory Markers and Clinical Characteristics

#### 3.2.1. Findings in the Full Sample

The findings for the linear regression model in the full sample, determining whether clinical characteristics (BMI, EDE-Q and BDI-II) were associated with cytokine concentrations after correction for age, ethnicity, smoking status and use of psychopharmacological medication, are presented in Figure 1B. The full results of the regression models in the total sample can be seen in Table 3. The residual spread remained non-normally distributed after the removal of outliers for many of the regression models.

#### 3.2.2. Findings in the AN sample

Figure 1C shows the findings for the linear regression model in the AN only sample, examining whether clinical characteristics (BMI, EDE-Q, BDI, duration of illness since diagnosis, probable PTSD) were associated with cytokine concentrations, whilst controlling for age, ethnicity, smoking status and use of psychopharmacological medications. Table 4 shows the full results of the regression models in the AN sample. BMI was positively associated with IP-10, with a moderate effect size, and IL-7, IL-12/IL-23–40, IL-16, with a large effect size. On the contrary, BMI was negatively related to IL-8, IL-15, IL-31, MCP-4, MIP-1α and MIP-1β, with a moderate effect size in this sample.

In terms of eating disorder psychopathology, a positive association was found with IFN-γ and MIP-3α, with a moderate effect size, and IL-12/IL-23p40, with a large effect size. However, eating disorder psychopathology was negatively associated with concentrations of IL-22, IL-27, MCP-1, MIP-1β and TNF-β, with a moderate effect size, and Eotaxin-3, with a large effect size. Illness duration (defined as years since diagnosis of AN) was positively associated with MIP-1β and negatively associated with nine inflammatory markers: IL-8, IL-10, IL-17α, IL-31, MDC, MIP-1α, MIP-3α (moderate effect size), IL-22 and IP-10 (large effect size).

Depressive symptomatology was positively and moderately related to IL-12/IL-23p40, IL-17α, IFN-γ and MIP-3α, and negatively related to IL-10 and IL-22, with a moderate effect size, and IL-27, with a large effect size. A probable diagnosis of PTSD was positively related to IL-12/IL-23p40 and MIP-3α, as well as IL-2, IL-4 and IL-7, all with a moderate effect size. On the other hand, PTSD was negatively and moderately associated with MIP-1α and MIP-1β.

## 4. Discussion

The present study quantified a broad range of inflammatory markers in people with acute AN and those recovered from AN, in comparison to HCs. Overall, the majority of inflammatory marker concentrations were unaltered in the AN and rec-AN groups, which aligns with the findings of our previous study [10]. However, there were also specific findings that contrast with the pro-inflammatory profile reported in our recent review [3] and with previous findings from our group where increased concentrations of pro-inflammatory cytokines, such as IL-6, IL-1β and TNF-α, were found in AN [9,10].

The present study revealed that, relative to HCs, concentrations of IL-7 and IL-12/IL-23p40 were reduced in AN and concentrations of Eotaxin-3, MDC, MIP-1α and TNF-α were reduced in rec-AN. When comparing to previous studies, there is converging evidence for reductions in IL-7 in AN [52], whereas, in contrast, concentrations of IL-12/IL-23p40 have been found to be similar across AN and HC [10] and concentrations of MIP-1α were similar across rec-AN, AN and HC in another study [11]. In both AN and rec-AN, MIP-1β was reduced compared to HCs, suggesting a potential trait marker of the illness. This is a novel finding; two previous studies have found no difference between AN and HCs in MIP-1β [10] nor between rec-AN, AN and HC [11]. There were also several differences in cytokines between the AN and rec-AN groups, including increased IL-16 and lowered IL-22 in the rec-AN group in comparison to the AN group. Moreover, the chemokines MIP-1α and Eotaxin-3 were lower in the rec-AN group compared to the AN group.

We also aimed to investigate the association between inflammatory markers and clinical characteristics of the disorder. Several associations emerged between the above-mentioned cytokines and anthropometric and clinical variables related to AN, such as BMI, illness duration and eating disorder psychopathology. In the full sample, associations between BMI and some inflammatory markers (e.g., GM-CSF, IL-8, IL-6, IL-16, IP-10) were mirrored by an inverse relationship with eating disorder psychopathology. However, this pattern of findings was not observed in AN.

BMI was positively related to IL-7 and IL-12/IL-23p40 in both the whole sample and the AN sample separately, which aligns with prior findings [10,52]. Additionally, IL-12/IL-23p40 was positively related to eating disorder psychopathology in the AN group. Both IL-7 and IL-12/IL-23p40 are pleiotropic cytokines that are integral for coordinating the pro-inflammatory response and stimulating the release of pro-inflammatory cytokines (e.g., TNF-α; [53,54,55]) and are associated with body weight [56,57,58], which is a likely explanation for the reductions found in the AN group. IL-16 was positively associated with BMI in the full sample (aligning with previous findings; [11]) and was elevated in the rec-AN group compared to AN, and IL-22 was negatively related to BMI and was lower in the rec-AN group compared to the AN group. Taken together, these findings may be indicative of recent weight gain in rec-AN, although further research would be needed to investigate this. However, IL-22 was also negatively associated with eating disorder psychopathology, depression and illness duration in the AN sample, indicating that a greater severity of psychiatric illness is linked to lower IL-22. Therefore, it is possible that a reduction in rec-AN is a trait of the illness, whereas in the acute state, low body weight has a greater impact on IL-22 than the psychological aspects of the illness and so a reduction in AN was not observed.

Chemokines such as MIP-1β have received relatively less attention in the psychiatric literature, although a recent meta-analysis found that MIP-1β is lower in depressed patients [30]. In the present study, concentrations of MIP-1β were lower in both the AN and rec-AN groups compared to HCs. MIP-1β was negatively related to both BMI and eating disorder psychopathology, as well as trauma, in the AN sample. Therefore, it is possible that the severity of AN and trauma may be contributing to the observed reduction in MIP-1β in the AN group. However, concentrations of MIP-1β were not related to any clinical characteristics in the full sample. Moreover, whilst Eotaxin-3 was reduced in the rec-AN group compared to the AN and HC groups, it was unrelated to any clinical or anthropometric variable in the full sample, and we were unable to explore these relationships further in the rec-AN group separately, due to the small sample size. There is ambiguity in why the rec-AN group showed reduced concentrations of MIP-1β and Eotaxin-3, although there are clearly complex relationships between symptoms of eating disorders and inflammatory markers in AN and particularly in rec-AN.

There were several associations between inflammatory markers and comorbid symptoms, such as depression and a probable diagnosis of PTSD. For example, depression was negatively related to IL-7 in the full sample, and PTSD was positively related to IL-7 in the AN sample. This suggests that PTSD is associated with increases in IL-7 in the AN sample, which is consistent with previous findings of elevations in IL-7 in individuals with PTSD and autoimmune disease, compared to autoimmune disease alone [59]. Moreover, IL-12/IL-23p40 was positively associated with depression and PTSD in the AN group. However as aforementioned, low BMI is likely to be the driver behind the observed reductions in IL-7 and IL-12/IL-23p40 in AN compared to HC. While depression and PTSD would usually increase concentrations of these pro-inflammatory cytokines [59,60,61], from our findings, it can be speculated that these effects are mitigated by low weight in AN (i.e., associated with a reduction in pro-inflammatory concentrations). Indeed, a previous study found aberrations in the concentrations of several inflammatory markers in AN, which was related to low BMI [11].

Overall, it is likely that extremes in BMI have a greater effect on many inflammatory markers (e.g., IL-7, IL-12/IL-23p40, IL-22) than comorbid psychiatric symptoms and/or eating disorder psychopathology. However, it is still unclear why some inflammatory markers, such as TNF-α and MIP-1α, were lower in rec-AN compared to HC. It is possible that there is an unmeasured variable driving these differences. For example, it is possible that psychological resilience to stress (imparted by psychological therapies) is present in rec-AN, resulting in a lesser inflammatory response [62,63,64]. Alternatively, it is possible that those with rec-AN have a more optimal dietary pattern as a result of their eating disorder; findings from a study in patients with heart failure found lower TNF-α levels in patients with diets lower in saturated fats [65]. Overall, it is apparent that rec-AN may be an interesting group, immunologically speaking, for future research to investigate further.

### Strengths and Limitations

The main strength of this study is the inclusion of a sample of individuals recovered from AN, which adds to the limited existing literature examining whether inflammatory aberrations in AN are state or trait aspects of the disorder. The study also measured a broad range of cytokines and chemokines, including Th17 cytokines, whilst statistically controlling for potentially confounding variables that are known to influence cytokine concentrations, such as age, ethnicity, smoking status and use of psychopharmacological medication. The discrepancies between the findings of the present study and the overall literature may be in part due to this statistical control of these variables. Additionally, as AN and autoimmune conditions have overlapping epidemiology [26], our stringent exclusion criteria of the presence of an autoimmune condition may have contributed to the inconsistency of our findings compared to previous studies; the majority have not considered autoimmune conditions (with the exception of one; [10]). Future research should consider including an additional group of participants with AN and a comorbid autoimmune condition to explore this and the role of inflammatory markers in the relationship between autoimmune conditions and eating disorders

There were several limitations to this cross-sectional study. Due to time and resource limitations, a small rec-AN sample was recruited, and it is likely that this aspect of the study is underpowered. Moreover, the groups were heterogeneous in terms of age and ethnicity, which, as aforementioned, can influence concentrations of inflammatory markers [47,48], although we statistically controlled for this influence. Additionally, the AN group varied in clinical characteristics, such as eating disorder severity, illness duration, weight and use of medication. Similarly, we included AN participants with both AN-restricting and AN binge/eating purging type. While eating disorder behaviours associated with AN-BP may be associated with alterations in pro-inflammatory markers via effects on the gut microbiome [66], only nine (out of 56) AN participants in our sample were diagnosed with AN-BP. Stage-based models of AN suggest that inflammation as an aspect of neuroprogression can vary depending on the stage of the illness [33]. Therefore, future studies may benefit from clearly delineating the sample on an aspect of severity, as this may influence inflammatory marker concentration. However, there is no agreed definition of long-term, or “severe-enduring” AN; therefore, this aspect of severity is difficult to isolate. For example, illness duration is often used as a marker of severity, although in this study it showed no relation to inflammatory marker concentrations in the AN group. Relatedly, we did not collect data on duration of recovery in rec-AN, which may be of interest to explore in future research exploring the association between rec-AN and inflammatory markers.

Most of the cytokine levels did not differ significantly between people with AN and HCs. It is unclear whether this finding is indicative of a relatively preserved immune system in AN. Such a speculation would need to be substantiated by prospective studies testing how people with AN cope with various infectious diseases. In this regard, we would additionally like to emphasise that we did not investigate important cellular parameters of the immune system such as electrolytes and the leukocyte count [67], and more specifically the concentration of Th1, Th2, Th17 and Tregs [68], which play key roles in the immune defence against infectious diseases. Indeed, it has been found previously that individuals with AN have lowered levels of monocytes, lymphocytes and neutrophils [69].

Finally, it is important to consider that many cytokines are present in the peripheral blood at low concentrations and may be undetectable due to poor assay sensitivity [70]. In the present study, there were several inflammatory markers below the detection limit (IL-1α, IL-1β, IL-5, IL-21 and IL-23), and so the data were excluded from the main analyses. There is no clear agreed method for handling non-detectable data in terms of percentage thresholds to impute the data and the method of imputing data. The present study was liberal in imputing data when more than 50% was present. It is important to consider that this chosen method may influence the findings in over- or under-estimating group differences.

## 5. Conclusions

This cross-sectional study investigated differences between three groups of individuals on a broad range of inflammatory markers: individuals with AN, individuals recovered from AN and healthy individuals with no psychiatric disorders. Furthermore, it aimed to ascertain whether alterations were related to clinical characteristics of the disorder and whether a distinct inflammatory profile is a state or trait marker of the disorder. There were no pro-inflammatory cytokines elevated in AN or rec-AN compared to HC in this study, contrasting with previous findings. In fact, the findings from the present study suggested lower concentrations of the cytokines IL-7 and IL-12/IL-23p40 in AN (potential state markers) and lower concentrations of TNF-α in rec-AN. Based on the findings of this study, reductions in the chemokine MIP-1β are a potential trait marker of the illness. Overall, the inflammatory profile of the acute AN group was mainly unremarkable, which may reflect the overall preserved immune function in this patient group, despite extensive dietary restriction and weight loss.

## Figures and Tables

**Figure 1 jpm-11-00814-f001:**
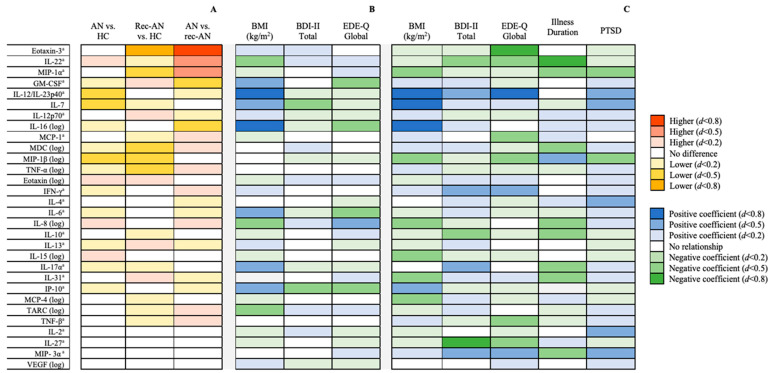
(**A**) Between-group differences in inflammatory marker concentrations after controlling for age, ethnicity, smoking status and use of psychopharmacological medications, stratified by effect size. (**B**) Associations between clinical characteristics and log10-transformed inflammatory markers in the whole sample (*n* = 131) after controlling for covariates, stratified by effect size. (**C**) Associations between clinical characteristics and log10-transformed inflammatory markers in the anorexia nervosa group (*n* = 56) after controlling for covariates, stratified by effect size. ^a^ Analysed by nonparametric analysis of covariance (Quade’s method). Abbreviations: AN = anorexia nervosa; BDI = Beck Depression Inventory; EDE-Q = Eating Disorders Examination Questionnaire; HC = healthy control; IFN = interferon; IL = interleukin; IP = interferon γ-induced protein; MCP = monocyte chemoattractant protein; MDC = macrophage-derived chemokine; MIP = macrophage inflammatory protein; rec-AN = recovered AN; TARC = thymus and activation-regulated chemokine; TNF = tumour necrosis factor; VEGF = vascular endothelial growth factor.

**Table 1 jpm-11-00814-t001:** Demographic and clinical characteristics for HC, AN and rec-AN groups.

	Healthy Controls (*n* = 51)	Current Anorexia Nervosa (*n* = 56)	Recovered Anorexia Nervosa (*n* = 24)	*p*-Value (Cohen’s *d*)
Demographic Characteristics				
Age (years) (mean ± SD)	23.5 ± 3.9	26.6 ± 8.1	26.6 ± 6.1	0.032 * (0.47)
*Ethnicity*				<0.001 ** (1.11)
White	27	50	22	
Ethnic minorities	24	6	2	
Current smoker (*n*)	7	6	3	0.892 (0.09)
Uses illicit drugs (*n*)	4	4	3	0.719 (0.15)
BMI (kg/m^2^) (mean ± SD) ^a^	21.3 ± 1.8	16.0 ± 1.2	20.8 ± 2.0	<0.001 ** (3.28)
Body fat (%) (mean ± SD)	24.0 ± 0.7	11.7 ± 0.7	22.4 ± 1.1	<0.001 ** (2.25)
Clinical Characteristics				
*AN subtype*				0.037 * (0.51)
Restricting	-	47	24	
Binge-eating/purging	-	9	0	
Duration of diagnosis, years (mean ± SD)	-	7.1 ± 8.0	6.3 ± 6.1 ^c^	0.841 (0.06)
Use of antidepressant medication (*n*)	-	26	6	<0.001 ** (1.36)
Use of antipsychotic medication (*n*)	-	6	2	0.061 (0.44)
Presence of probable PTSD (*n*)	1	23	5	<0.001 ** (0.85)
EDE-Q Global (mean ± SD) ^b^	0.6 ± 0.6	3.9 ± 1.1	1.0 ± 0.8	<0.001 ** (3.50)
BDI-II Total (mean ± SD) ^b^	4.0 ± 4.5	30.6 ± 11.4	11.6 ± 9.5	<0.001 ** (2.76)
DASS-21 Total (mean ± SD) ^b^	8.5 ± 9.0	60.8 ± 27.0	26.8 ± 22.7	<0.001 ** (2.31)

Notes: ^a^ Log transformed; ^b^ Non-parametric test used; ^c^ Data from *n* = 4 respondents. * significant at the *p* < 0.05 threshold, ** significant at the *p* < 0.01 threshold.

**Table 2 jpm-11-00814-t002:** Raw mean (±standard deviation) and adjusted marginal mean (±standard error) concentrations of each cytokine with the statistical significance of the group comparison from the ANCOVA models, controlling for age, ethnicity, use of pharmacological medication and smoking status.

Inflammatory Marker	Healthy Controls (*n* = 51)		Anorexia Nervosa (*n* = 56)		Recovered AN (*n* = 24)		*p*-Value (Cohen’s *d*)			
	M ± SD (pg/mL)	Adjusted M ± SE	M ± SD (pg/mL)	Adjusted M ± SE	M ± SD (pg/mL)	Adjusted M ± SE	Total Model	HC vs. AN.	HC vs. Rec-AN	AN vs. Rec-AN
Eotaxin (log)	174.37 ± 10.05	2.24 ± 0.03	236.31 ± 13.36	2.32 ± 0.03	230.48 ± 11.76	2.32 ± 0.04	0.084 *(0.40)*	0.049 * *(0.36)*	0.052 *(0.35)*	0.861 *(0.03)*
Eotaxin-3 ^a^	16.48 ± 1.51	5.68 ± 4.74	20.52 ± 5.99	8.95 ± 4.53	7.78 ± 0.62	−32.95 ± 6.91	**<0.001 ** *(0.90)***	0.847 *(0.03)*	**<0.001 ** *(0.81)***	**<0.001 ** *(0.85)***
GM-CSF ^a^	0.12 ± 0.02	2.02 ± 4.96	0.08 ± 0.01	−8.59 ± 4.74	0.14 ± 0.02	15.74 ± 7.24	0.019 * ***(0.51)***	0.124 *(0.27)*	0.120 *(0.28)*	**0.006 ** *(0.50)***
IFN-γ ^a^	22.45 ± 9.63	6.48 ± 5.18	5.69 ± 1.26	−6.54 ± 4.94	4.96 ± 1.18	1.51 ± 7.55	0.190 *(0.33)*	0.071 *(0.32)*	0.588 *(0.09)*	0.374 *(0.16)*
IL-2 ^a^	0.26 ± 0.04	0.29 ± 5.31	0.32 ± 0.06	−2.75 ± 5.06	0.35 ± 0.09	5.81 ± 7.73	0.650 *(0.17)*	0.679 *(0.06)*	0.557 *(0.11)*	0.356 *(0.17)*
IL-4 ^a^	0.03 ± 0.01	−0.41 ± 5.12	0.02 ± 0.003	−3.61 ± 4.89	0.03 ± 0.01	9.30 ± 7.46	0.352 *(0.26)*	0.652 *(0.09)*	0.285 *(0.19)*	0.150 *(0.26)*
IL-6 ^a^	1.50 ± 0.56	5.33 ± 4.91	0.38 ± 0.07	−9.27 ± 4.68	0.42 ± 0.04	10.30 ± 7.15	0.033 * *(0.47)*	0.033 * *(0.38)*	0.568 *(0.11)*	0.025 * *(0.40)*
IL-7	19.12 ± 0.83	18.4 ± 0.9	14.23 ± 0.63	14.8 ± 0.8	15.39 ± 0.94	15.7 ± 1.1	0.019 * ***(0.51)***	**0.005 ** *(0.51)***	0.058 *(0.35)*	0.468 *(0.13)*
IL-8 (log)	8.89 ± 0.46	0.93 ± 0.03	11.15 ± 0.53	1.02 ± 0.02	9.38 ± 0.71	0.94 ± 0.03	0.049 * *(0.44)*	0.036 * *(0.38)*	0.928 *(0.02)*	0.046 * *(0.36)*
IL-10 ^a^	0.41 ± 0.08	−0.09 ± 5.27	0.39 ± 0.06	4.76 ± 5.03	0.25 ± 0.03	−10.90 ± 7.68	0.237 *(0.30)*	0.507 *(0.11)*	0.248 *(0.20)*	0.091 *(0.30)*
IL-12p70 ^a^	0.23 ± 0.05	−1.77 ± 5.12	0.23 ± 0.04	−5.92 ± 4.89	0.29 ± 0.04	17.57 ± 7.47	0.031 * *(0.47)*	0.560 *(0.11)*	0.035 * *(0.38)*	**0.010 **** *(0.46)*
IL-12/IL-23p40 ^a^	124.82 ± 5.58	9.13 ± 4.85	94.82 ± 9.21	−10.54 ± 4.63	114.30 ± 11.04	5.20 ± 7.07	**0.011 * *(0.54)***	**0.004 ** *(0.52)***	0.647 *(0.09)*	0.065 *(0.33)*
IL-13 ^a^	1.44 ± 0.19	2.28 ± 5.13	1.47 ± 0.45	−8.15 ± 4.90	1.74 ± 0.37	14.18 ± 7.48	0.041 * *(0.45)*	0.144 *(0.26)*	0.192 *(0.23)*	**0.014 *** *(0.44)*
IL-15 (log)	2.67 ± 0.12	0.39 ± 0.02	2.67 ± 0.09	0.43 ± 0.02	2.65 ± 0.09	0.42 ± 0.02	0.401 *(0.25)*	0.191 *(0.24)*	0.319 *(0.18)*	0.834 *(0.04)*
IL-16 (log)	185.90 ± 8.45	2.26 ± 0.02	165.89 ± 7.39	2.18 ± 0.02	202.33 ± 11.61	2.29 ± 0.03	**0.004 ** *(0.62)***	0.019 * *(0.43)*	0.442 *(0.14)*	**0.002 ** *(0.58)***
IL-17α ^a^	2.62 ± 0.16	10.87 ± 4.96	1.78 ± 0.14	−6.84 ± 4.73	1.78 ± 0.14	−7.16 ± 7.23	0.022 * ***(0.50)***	0.011 * *(0.46)*	0.042 * *(0.36)*	0.970 *(0.01)*
IL-22 ^a^	7.73 ± 3.41	−1.84 ± 5.10	2.26 ± 0.30	8.64 ± 4.87	1.28 ± 0.34	−16.25 ± 7.44	0.020 * ***(0.50)***	0.140 *(0.26)*	0.113 *(0.29)*	**0.006 ** *(0.50)***
IL-27 ^a^	1054.03 ± 77.59	−2.86 ± 5.26	1043.02 ± 52.00	1.05 ± 5.02	1032.73 ± 60.15	3.63 ± 7.67	0.755 *(0.13)*	0.591 *(0.09)*	0.486 *(0.13)*	0.779 *(0.06)*
IL-31 ^a^	0.11 ± 0.02	−3.73 ± 5.07	0.11 ± 0.02	−0.70 ± 4.84	0.15 ± 0.03	9.57 ± 7.39	0.329 *(0.26)*	0.666 *(0.06)*	0.140 *(0.26)*	0.247 *(0.20)*
IP-10 ^a^	300.77 ± 24.35	3.60 ± 5.16	246.45 ± 20.06	−6.75 ± 4.93	294.77 ± 32.19	8.09 ± 7.52	0.176 *(0.33)*	0.149 *(0.26)*	0.624 *(0.09)*	0.101 *(0.29)*
MCP-1 ^a^	249.80 ± 12.51	5.22 ± 5.05	230.27 ± 7.98	3.43 ± 4.82	200.72 ± 10.86	−19.10 ± 7.36	0.018 * ***(0.51)***	0.797 *(0.06)*	**0.007 **** *(0.48)*	**0.012 *** *(0.45)*
MCP-4 (log)	103.75 ± 5.27	2.00 ± 0.03	107.39 ± 6.89	2.00 ± 0.03	93.88 ± 8.40	1.90 ± 0.04	0.358 *(0.26)*	0.619 *(0.09)*	0.163 *(0.26)*	0.297 *(0.19)*
MDC (log)	1600.36 ± 75.93	3.17 ± 0.02	1340.28 ± 62.2	3.10 ± 0.02	1042.02 ± 55.39	3.0 ± 0.03	**0.001 ** *(0.69)***	0.107 *(0.29)*	**0.001 ** *(0.68)***	**0.012 *** *(0.46)*
MIP-1α ^a^	184.96 ± 20.73	8.19 ± 4.84	136.50 ± 16.01	4.63 ± 4.62	14.13 ± 1.00	−28.21 ± 7.06	**<0.001 ** *(0.79)***	0.596 *(0.09)*	**<0.001 ** *(0.75)***	**<0.001 ** *(0.69)***
MIP-1β (log)	129.63 ± 8.89	2.06 ± 0.03	91.59 ± 5.30	1.93 ± 0.03	87.12 ± 5.66	1.92 ± 0.04	**0.006 ** *(0.59)***	**0.004 ** *(0.52)***	**0.005 ** *(0.51)***	0.816 *(0.04)*
MIP- 3α ^a^	11.39 ± 2.85	1.08 ± 4.76	5.36 ± 0.38	−0.14 ± 4.58	4.88 ± 0.38	−1.96 ± 6.93	0.936 *(0.06)*	0.854 *(0.03)*	0.718 *(0.06)*	0.827 *(0.04)*
TARC (log)	305.12 ± 21.68	2.45 ± 0.04	352.90 ± 29.26	2.47 ± 0.04	260.95 ± 39.35	2.34 ± 0.05	0.083 *(0.40)*	0.737 *(0.06)*	0.096 *(0.30)*	0.031 * *(0.39)*
TNF-α (log)	1.96 ± 0.10	0.27 ± 0.02	1.70 ± 0.07	0.21 ± 0.02	1.37 ± 0.05	0.13 ± 0.03	**0.003 ** *(0.63)***	0.097 *(0.30)*	**0.001 ** *(0.63)***	0.033 * *(0.39)*
TNF-β ^a^	0.15 ± 0.02	3.47 ± 5.25	0.14 ± 0.01	3.00 ± 5.01	0.09 ± 0.02	−14.37 ± 7.66	0.120 *(0.37)*	0.949 *(0.01)*	0.057 *(0.34)*	0.060 *(0.33)*
VEGF (log)	102.37 ± 8.68	1.91 ± 0.05	81.46 ± 6.72	1.87 ± 0.04	98.54 ± 14.34	1.91 ± 0.06	0.737 *(0.14)*	0.531 *(0.11)*	0.964 *(0.01)*	0.501 *(0.13)*

^a^ Analysed by nonparametric analysis of covariance (Quade’s method), * significant at the *p* < 0.05 threshold, ** significant at the *p* < 0.01 threshold. Emboldened *p*-values are significant after applying Benjamini–Hochberg FDR correction for multiple comparisons, and emboldened Cohen’s *d* values indicate moderate or large effect sizes. Abbreviations: AN= anorexia nervosa; IFN = interferon; IL = interleukin; IP = interferon γ-induced protein; M = mean value; MCP = monocyte chemoattractant protein; MDC = macrophage-derived chemokine; MIP = macrophage inflammatory protein; SE = standard error; TARC = thymus and activation-regulated chemokine; TNF = tumour necrosis factor; VEGF = vascular endothelial growth factor.

**Table 3 jpm-11-00814-t003:** Associations between clinical characteristics and inflammatory markers in the whole sample (*n* = 131) after controlling for age, ethnicity, smoking status and use of psychopharmacological medication.

Inflammatory Marker		Body Mass Index		EDE-Q Global Score		BDI Score	
	N Residual Outliers	β Coefficient (95% CIs)	*p* (Cohen’s *d)*	β Coefficient (95% CIs)	*p* (Cohen’s *d)*	β Coefficient (95% CIs)	*p* (Cohen’s *d)*
Eotaxin	0	−0.18 (−0.02, −0.001)	0.040 * (−*0.47)*	0.12 (−0.01, 0.03)	0.209 *(0.35)*	0.14 (−0.001, 0.004)	0.132 *(0.39)*
Eotaxin-3	1	−0.15 (−0.03, 0.004)	0.130 (−*0.41)*	0.04 (−0.02, 0.04)	0.708 *(0.18)*	0.05 (−0.003, 0.01)	0.612 *(0.20)*
GM-CSF	0	0.22 (0.04, 0.47)	0.023 ***(0.56)***	−0.20 (−0.76, −0.001)	0.049 * ***(−0.52)***	−0.02 (−0.05, 0.04)	0.882 *(−0.14)*
IFN-γ	8	0.14 (−0.01, 0.04)	0.235 *(0.39)*	0.01 (−0.03, 0.04)	0.925 *(0.12)*	0.02 (−0.004, 0.01)	0.846 *(0.14)*
IL-2	9	−0.19 (−0.26, 0.004)	0.058 *(−0.49)*	−0.18 (−0.10, 0.01)	0.101 *(−0.47)*	−0.14 (−0.01, 0.002)	0.198 *(0.39)*
IL-4	0	−0.03 (−0.26, 0.20)	0.786 *(−0.16)*	−0.06 (−0.51, 0.29)	0.577 *(−0.22)*	0.01 (−0.05, 0.05)	0.935 *(0.12)*
IL-6	6	0.26 (0.01, 0.04)	0.007 ** ***(0.65)***	−0.27 (−0.07, −0.01)	0.007 ** ***(−0.68)***	−0.14 (−0.01, 0.001)	0.183 *(−0.39)*
IL-7	2	0.30 (0.01, 0.02)	**0.001** ** ***(0.75)***	−0.18 (−0.03, 0.001)	0.064 *(−0.47)*	−0.20 (−0.004, −0.001)	0.043 * ***(−0.52)***
IL-8	0	−0.29 (−0.03, −0.01)	**0.002** ** ***(−0.72)***	0.23 (0.003, 0.04)	0.021 * ***(0.58)***	0.15 (−0.001, 0.004)	0.136 *(0.41)*
IL-10	1	−0.14 (−0.03, 0.01)	0.169 *(−0.38)*	0.06 (−0.02, 0.04)	0.587 *(0.22)*	−0.02 (−0.004, 0.003)	0.844 *(−0.14)*
IL-12p70	0	0.14 (−0.05, 0.31)	0.147 *(0.39)*	−0.08 (−0.44, 0.19)	0.429 *(−0.26)*	−0.05 (−0.05, 0.03)	0.656 *(−0.20)*
IL-12/IL-23p40	3	0.38 (0.01, 0.04)	**<0.001** ** ***(0.95)***	−0.11 (−0.03, 0.01)	0.251 *(−0.32)*	−0.15 (−0.01, 0.001)	0.127 *(−0.41)*
IL-13	0	0.11 (−0.10, 0.36)	0.264 *(0.32)*	−0.13 (−0.66, 0.14)	0.194 *(−0.37)*	−0.04 (−0.06, 0.04)	0.701 *(−0.18)*
IL-15	2	−0.16 (−0.01, 0.001)	0.105 *(−0.43)*	0.04 (−0.01, 0.01)	0.717 *(0.18)*	0.04 (−0.001, 0.002)	0.677 *(0.18)*
IL-16	0	0.37 (0.01, 0.03)	**<0.001** ** ***(0.93)***	−0.26 (−0.04, −0.01)	0.011 * ***(−0.65)***	−0.12 (−0.003, 0.001)	0.263 *(−0.35)*
IL-17α	2	0.20 (0.001, 0.03)	0.030 * ***(0.52)***	−0.15 (−0.04, 0.01)	0.139 *(−0.41)*	−0.08 (−0.004, 0.002)	0.399 *(−0.26)*
IL-22	4	−0.24 (−0.07, −0.01)	0.015 * ***(−0.61)***	0.19 (−0.003, 0.10)	0.065 *(0.49)*	0.13 (−0.002, 0.01)	0.222 *(0.37)*
IL-27	1	−0.11 (−0.02, 0.01)	0.254 *(−0.32)*	−0.06 (−0.03, 0.01)	0.599 *(−0.22)*	−0.04 (−0.003, 0.002)	0.675 *(−0.18)*
IL-31	0	0.03 (−0.22, 0.29)	0.800 *(0.16)*	0.05 (−0.34, 0.53)	0.670 *(0.20)*	0.0003 (−0.05, 0.05)	0.998 *(0.10)*
IP-10	2	0.23 (0.002, 0.03)	0.021 * ***(0.58)***	−0.25 (−0.05, −0.01)	0.017 * ***(−0.63)***	−0.24 (−0.01, −0.001)	0.023* ***(−0.61)***
MCP-1	1	−0.07 (−0.01, 0.01)	0.482 *(−0.24)*	−0.02 (−0.02, 0.01)	0.831 *(−0.14)*	−0.01 (−0.002, 0.002)	0.911 *(−0.12)*
MCP-4	0	−0.18 (−0.02, 0.001)	0.069 *(−0.47)*	−0.03 (−0.02, 0.02)	0.784 *(−0.16)*	0.04 (−0.002, 0.003)	0.689 *(0.18)*
MDC	0	−0.01 (−0.01, 0.01)	0.955 *(−0.12)*	0.02 (−0.02, 0.02)	0.863 *(0.14)*	0.06 (−0.002, 0.003)	0.585 *(0.22)*
MIP-1α	0	−0.15 (−0.08, 0.01)	0.117 *-(0.41)*	0.11 (−0.04, 0.12)	0.299 *(0.32)*	0.03 (−0.01, 0.01)	0.756 *(0.16)*
MIP-1β	0	0.03 (−0.01, 0.01)	0.747 *(0.16)*	−0.17 (−0.04, 0.003)	0.092 *(−0.45)*	−0.19 (−0.01, −0.001)	0.069 *(−0.49)*
MIP- 3α	7	0.03 (−0.01, 0.02)	0.776 *(0.16)*	0.06 (−0.02, 0.03)	0.531 *(0.22)*	−0.01 (−0.003, 0.003)	0.930 *(−0.12)*
TARC	0	−0.21 (−0.03, −0.002)	0.028 * ***(−0.54)***	0.10 (−0.01, 0.04)	0.346 *(0.30)*	0.19 (0.001, 0.006)	0.070 *(0.49)*
TNF-α	0	0.01 (−0.01, 0.01)	0.943 *(0.12)*	−0.01 (−0.02, 0.02)	0.959 *(−0.12)*	−0.04 (−0.002, 0.002)	0.673 *(−0.18)*
TNF-β	0	−0.01 (−0.19, 0.17)	0.923 *(−0.12)*	−0.004 (−0.32, 0.31)	0.971 *(−0.11)*	0.03 (−0.03, 0.04)	0.753 *(0.16)*
VEGF	0	0.08 (−0.01, 0.03)	0.402 *(0.26)*	−0.05 (−0.04, 0.02)	0.641 *(−0.20)*	−0.09 (−0.01, 0.002)	0.401 *(−0.28)*

* significant at the *p* < 0.05 threshold, ** significant at the *p* < 0.01 threshold. Emboldened *p*-values are significant after applying Benjamini–Hochberg FDR correction for multiple comparisons, and emboldened Cohen’s *d* values indicate moderate or large effect sizes. Abbreviations: BDI = Beck Depression Inventory; CI = confidence interval; EDE-Q = Eating Disorders Examination Questionnaire; IFN = interferon; IL = interleukin; IP = interferon γ-induced protein; MCP = monocyte chemoattractant protein; MDC = macrophage-derived chemokine; MIP = macrophage inflammatory protein; TARC = thymus and activation-regulated chemokine; TNF = tumor necrosis factor; VEGF = vascular endothelial growth factor. In the full sample, BMI was positively associated with concentrations of GM-CSF, IL-6, IL-7, IL-17α and IP-10, with a moderate effect size, and IL-16 and IL-12/IL-23p40 with a large effect size. BMI was negatively associated with concentrations of IL-8, IL-22 and TARC, with a moderate effect size. Eating disorder psychopathology was positively associated with concentrations of IL-8 and negatively associated with concentrations of GM-CSF, IL-6, IL-16 and IP-10, all with moderate effect sizes. Depression symptoms, as measured by the BDI-II, were negatively and moderately associated with concentrations of IL-7 and IP-10.

**Table 4 jpm-11-00814-t004:** Associations between clinical characteristics and inflammatory markers in the anorexia nervosa group (*n* = 56) after controlling for age, ethnicity, smoking status and use of psychopharmacological medication.

Inflammatory Marker	N Residual Outliers	Body Mass Index		EDE-Q Global Score		BDI Score		Illness Duration		Probable PTSD	
		β Coefficient (95% CIs)	*p* (Cohen’s *d)*	β Coefficient (95% CIs)	*p* (Cohen’s *d)*	β Coefficient (95% CIs)	*p* (Cohen’s *d)*	β Coefficient (95% CIs)	*p* (Cohen’s *d)*	β Coefficient (95% CIs)	*p* (Cohen’s *d)*
Eotaxin	0	−0.15 (−0.06, 0.02)	0.300 *(−0.41)*	0.07 (−0.03, 0.05)	0.608 *(0.24)*	0.08 (−0.003, 0.01)	0.575 *(0.26)*	−0.09 (−0.01, 0.01)	0.683 *(−0.14)*	0.07 (−0.08, 0.13)	0.642 *(0.24)*
Eotaxin-3	2	−0.08 (−0.07, 0.04)	0.614 *(−0.26)*	−0.34 (−0.12, −0.01)	0.018 * ***(−0.85)***	−0.12 (−0.01, 0.004)	0.431 *(−0.35)*	0.03 (−0.01, 0.01)	0.911 *(0.16)*	−0.13 (−0.21, 0.08)	0.383 *(−0.37)*
GM-CSF	0	0.18 (−0.33, 1.43)	0.218 *(0.47)*	−0.08 (−1.18, 0.65)	0.565 *(−0.26)*	0.19 (−0.03, 0.16)	0.192 *(0.49)*	0.02 (−0.21, 0.22)	0.945 *(0.14)*	0.08 (−1.64, 2.87)	0.585 *(0.26)*
IFN-γ	0	0.18 (−0.03, 0.12)	0.241 *(0.47)*	0.25 (−0.01, 0.14)	0.070 ***(0.63)***	0.30 (0.001, 0.02)	0.049 * ***(0.75)***	−0.01 (−0.02, 0.02)	0.956 *(−0.12)*	0.15 (−0.09, 0.30)	0.300 *(0.41)*
IL-2	2	−0.13 (−0.18, 0.07)	0.406 *(−0.37)*	−0.10 (−0.17, 0.09)	0.514 *(−0.30)*	−0.01 (−0.01, 0.01)	0.956 *(−0.12)*	0.01 (−0.03, 0.03)	0.974 *(0.12)*	0.23 (−0.06, 0.56)	0.112 ***(0.58)***
IL-4	0	0.004 (−0.85, 0.87)	0.980 *(0.11)*	−0.11 (−1.23, 0.53)	0.430 *(−0.32)*	0.05 (−0.08, 0.11)	0.754 *(0.20)*	0.12 (−0.15, 0.26)	0.613 *(0.35)*	0.24 (−0.32, 3.91)	0.095 ***(0.61)***
IL-6	1	−0.08 (−0.07, 0.04)	0.583 *(−0.26)*	−0.10 (−0.08, 0.03)	0.448 *(−0.30)*	0.12 (−0.003, 0.01)	0.392 *(0.35)*	−0.12 (−0.01, 0.01)	0.930 *(−0.35)*	0.15 (−0.06, 0.21)	0.265 *(0.41)*
IL-7	1	0.34 (0.01, 0.08)	0.021 * ***(0.85)***	0.12 (−0.02, 0.05)	0.415 *(0.35)*	0.09 (−0.003, 0.01)	0.548 *(0.28)*	−0.10 (−0.01, 0.01)	0.673 *(−0.30)*	0.24 (−0.02, 0.16)	0.100 ***(0.61)***
IL-8	0	−0.29 (−0.07, 0.001)	0.044 ***(−0.72)***	0.001 (−0.04, 0.04)	0.994 *(0.10)*	−0.06 (−0.01, 0.003)	0.684 *(−0.22)*	−0.27 (−0.01, 0.003)	0.225 ***(−0.68)***	0.09 (−0.07, 0.12)	0.543 *(0.28)*
IL-10	0	−0.05 (−0.08, 0.06)	0.738 *(−0.20)*	−0.18 (−0.11, 0.03)	0.205 *(−0.47)*	−0.27 (−0.01, 0.001)	0.070 ***(−0.68)***	−0.26 (−0.03, 0.01)	0.252 ***(−0.65)***	−0.18 (−0.28, 0.07)	0.221 *(−0.47)*
IL-12p70	0	0.15 (−0.36, 1.10)	0.31 *(0.41)*	−0.09 (−1.00, 0.53)	0.548 *(−0.28)*	−0.06 (−0.10, 0.07)	0.702 *(−0.22)*	−0.10 (−0.22, 0.14)	0.670 *(0.30)*	0.19 (−0.58, 3.08)	0.177 *(0.49)*
IL-12/IL-23p40	1	0.45 (0.04, 0.14)	0.001 ** ***(1.15)***	0.35 (0.02, 0.13)	0.011 * ***(0.87)***	0.25 (−0.001, 0.01)	0.081 ***(0.63)***	0.03 (−0.01, 0.01)	0.899 *(0.16)*	0.23 (−0.03, 0.26)	0.108 ***(0.58)***
IL-13	0	−0.17 (−1.53, 0.39)	0.241 *(−0.45)*	0.002 (−1.00, 1.01)	0.987 *(0.10)*	0.11 (−0.07, 0.15)	0.452 *(0.32)*	−0.04 (−0.26, 0.21)	0.854 *(−0.18)*	−0.14 (−3.69, 1.17)	0.303 *(−0.39)*
IL-15	0	−0.26 (−0.05, 0.002)	0.068 ***(−0.65)***	−0.13 (−0.04, 0.01)	0.346 *(−0.37)*	−0.15 (−0.004, 0.001)	0.295 *(−0.41)*	0.02 (−0.01, 0.01)	0.917 *(0.14)*	−0.18 (−0.10, 0.02)	0.210 *(−0.47)*
IL-16	0	0.36 (0.01, 0.07)	0.008 ** ***(0.90)***	−0.04 (−0.04, 0.03)	0.773 *(−0.18)*	0.16 (−0.001, 0.01)	0.245 *(0.43)*	0.11 (−0.01, 0.01)	0.597 *(0.32)*	0.06 (−0.06, 0.09)	0.654 *(0.22)*
IL-17α	1	−0.004 (−0.06, 0.06)	0.979 *(−0.11)*	0.03 (−0.05, 0.07)	0.843 *(0.16)*	0.28 (−0.001, 0.12)	0.057 ***(0.70)***	−0.20 (−0.02, 0.01)	0.380 ***(−0.52)***	0.07 (−0.11, 0.18)	0.610 *(0.24)*
IL-22	0	−0.07 (−0.13, 0.08)	0.628 *(−0.24)*	−0.22 (−0.19, 0.02)	0.106 ***(−0.56)***	−0.22 (−0.02, 0.003)	0.135 ***(−0.56)***	−0.50 (−0.05, −0.004)	0.021* ***(−1.32)***	−0.10 (−0.35, 0.17)	0.490 *(−0.30)*
IL-27	0	−0.10 (−0.05, 0.02)	0.486 *(−0.30)*	−0.25 (−0.07, 0.003)	0.073 ***(−0.63)***	−0.32 (−0.01, 0.001)	0.030 * ***(−0.80)***	0.09 (−0.01, 0.01)	0.692 *(0.28)*	−0.08 (−0.12, 0.07)	0.573 *(−0.26)*
IL-31	0	−0.25 (−1.75, 0.15)	0.097 ***(−0.63)***	0.17 (−0.39, 1.59)	0.230 *(0.45)*	0.13 (−0.06, 0.15)	0.408 *(0.37)*	−0.30 (−0.38, 0.08)	0.193 ***(−0.75)***	0.18 (−0.88, 3.98)	0.206 *(0.47)*
IP-10	0	0.27 (−0.002, 0.09)	0.063 ***(0.68)***	−0.19 (−0.08, 0.02)	0.183 *(−0.49)*	−0.08 (−0.01, 0.004)	0.574 *(−0.26)*	−0.47 (−0.02, −0.001)	0.035 * ***(−1.22)***	−0.05 (−0.14, 0.10)	0.718 *(−0.20)*
MCP-1	0	0.11 (−0.02, 0.04)	0.422 *(0.32)*	−0.21 (−0.05, 0.01)	0.115 ***(−0.54)***	0.004 (−0.003, 0.003)	0.980 *(0.11)*	0.19 (−0.003, 0.01)	0.274 *(0.49)*	−0.01 (−0.07, 0.06)	0.948 *(−0.12)*
MCP-4	0	−0.22 (−0.08, 0.01)	0.135 ***(−0.56)***	−0.12 (−0.07, 0.03)	0.386 *(−0.35)*	0.12 (−0.003, 0.01)	0.419 *(0.35)*	−0.18 (−0.02, 0.01)	0.434 *(0.47)*	−0.10 (−0.16, 0.08)	0.503 *(−0.30)*
MDC	1	0.19 (−0.01, 0.05)	0.188 *(0.49)*	−0.05 (−0.04, 0.03)	0.736 *(−0.20)*	0.12 (−0.002, 0.01)	0.403 *(0.35)*	−0.29 (−0.01, 0.003)	0.219 ***(−0.72)***	0.10 (−0.06, 0.12)	0.489 *(0.30)*
MIP-1α	0	−0.20 (−0.27, 0.05)	0.181 ***(−0.52)***	−0.10 (−0.23, 0.11)	0.484 *(−0.30)*	−0.06 (−0.02, 0.01)	0.669 *(−0.22)*	−0.30 (−0.06, 0.01)	0.179 ***(−0.75)***	−0.23 (−0.75, 0.07)	0.098 ***(−0.58)***
MIP-1β	0	−0.29 (−0.08, −0.001)	0.042 * ***(−0.72)***	−0.20 (−0.07, 0.01)	0.161 ***(−0.52)***	−0.13 (−0.01, 0.002)	0.388 *(−0.37)*	0.30 (−0.003, 0.02)	0.182 ***(0.75)***	−0.22 (−0.18, 0.02)	0.120 ***(−0.56)***
MIP- 3α	1	0.08 (−0.03, 0.06)	0.564 *(0.26)*	0.29 (0.01, 0.10)	0.024 * ***(0.72)***	0.23 (−0.001, 0.01)	0.090 ***(0.58)***	−0.20 (−0.02, 0.01)	0.336 ***(−0.52)***	0.28 (0.01, 0.25)	0.034* ***(0.70)***
TARC	0	−0.13 (−0.08, 0.03)	0.354 *(−0.37)*	−0.04 (−0.07, 0.05)	0.755 *(−0.18)*	0.15 (−0.003, 0.01)	0.285 *(0.41)*	−0.16 (−0.02, 0.01)	0.457 *(−0.43)*	0.11 (−0.08, 0.20)	0.423 *(0.32)*
TNF-α	0	0.19 (−0.01, 0.05)	0.202 *(0.49)*	−0.10 (−0.04, 0.02)	0.493 *(−0.30)*	−0.09 (−0.01, 0.002)	0.538 *(−0.28)*	−0.12 (−0.01, 0.01)	0.608 *(−0.35)*	0.08 (−0.06, 0.10)	0.604 *(0.26)*
TNF-β	0	0.09 (−0.43, 0.80)	0.552 *(0.28)*	−0.25 (−1.17, 0.06)	0.074 ***(−0.63)***	−0.18 (−0.11, 0.03)	0.222 *(−0.47)*	−0.09 (−0.16, 0.11)	0.694 *(−0.28)*	0.09 (−1.07, 2.03)	0.541 *(0.28)*
VEGF	0	0.01 (−0.07, 0.07)	0.931 *(0.12)*	0.14 (−0.04, 0.11)	0.326 *(0.39)*	−0.03 (−0.01, 0.01)	0.835 *(−0.16)*	−0.001 (−0.02, 0.02)	0.996 *(−0.10)*	0.08 (−0.13, 0.22)	0.591 *(0.26)*

* significant at the *p* < 0.05 threshold before correction for multiple testing, ** significant at the *p* < 0.01 threshold before correction for multiple testing. Emboldened Cohen’s *d* values indicate moderate or large effect sizes. Abbreviations: BDI = Beck Depression Inventory; CI = confidence interval; EDE-Q = Eating Disorders Examination Questionnaire; IFN = interferon; IL = interleukin; IP = interferon γ-induced protein; MCP = monocyte chemoattractant protein; MDC = macrophage-derived chemokine; MIP = macrophage inflammatory protein; TARC = thymus and activation-regulated chemokine; TNF = tumour necrosis factor; VEGF = vascular endothelial growth factor.

## Data Availability

The data presented in this study are available on request from the corresponding author. The data are not publicly available as they are still being used for analysis and manuscript writing.

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
