# Peer review of "Reduced MIP-1β as a Trait Marker and Reduced IL-7 and IL-12 as State Markers of Anorexia Nervosa"

_jpm, 2021, doi:10.3390/jpm11080814_

Round 1

Reviewer 1 Report

In this study, the authors have used an array of  36 inflammatory markers/cytokines to find the differences in their expression level across three groups of individuals- (a) suffering from anorexia nervosa (AN), (b) recovered from AN, and (c) healthy controls.

They have also made an attempt to correlate the levels of inflammatory markers to various clinical characteristics like BMI, illness duration, eating disorder symptom severity and psychiatric comorbidities.

They used data from 131 participants (final number after inclusion and exclusion criteria) over the age of 18, divided into 3 above-mentioned groups.

The present study is a kind of extension of their previous work with a more controlled experimental design. The observations from the data have been discussed in detail.

However, there are a few minor concerns that the authors need to answer before the acceptance of the manuscript for publication.

1. Why were males subjects excluded from the study?
2. The authors mentioned that the levels of cytokines were not related to the illness duration in AN group. But it would be interesting to see whether or not their levels change in the recovered group with increasing time lapse. This will help in answering which cytokines get restored to the anormal levels upon recovering and which ones require to follow up monitoring.
3. In contrast of the previous reports, including their own, authors did not find elevated levels of pro-inflammatory cytokines in AN or recovered group.  Though authors have anticipated the role of subjects with autoimmune conditions in this regard, they could not reflect much on this part of the results. It would be great to have a fourth group of patients with AN+autoimmune condition to account for the unanswered questions.
Or authors can present a table showing inclusion and exclusion criteria of the previous studies that report the opposite trend for these pro-inflammatory cytokines to give readers a better insight in this direction.
4. ANCOVA has to be expanded at the first point of mention.
5. Typo- page 6, line 279... It should be 'different' in place of 'difference'. 

Author Response

  1. Why were males subjects excluded from the study?

As this is a new research area, and males with AN are difficult to recruit, we included only women in order to ensure that our sample was homogeneous. Additionally, there is research that indicates gender differences in inflammatory markers in both healthy individuals (e.g. Giraldo et al., 2008) and patients with depression (Wiener et al., 2018; Jha et al., 2018). If we included males, there would be few male participants per group, and sensitivity analyses or statistical control for gender would have been difficult due to issues with statistical power.

  1. The authors mentioned that the levels of cytokines were not related to the illness duration in AN group. But it would be interesting to see whether or not their levels change in the recovered group with increasing time lapse. This will help in answering which cytokines get restored to the anormal levels upon recovering and which ones require to follow up monitoring.

This is an interesting and valid point. Unfortunately, we do not have data available on recovery duration in our rec-AN group, and therefore these analyses would not be possible. However, we have added this in as a future direction in the following sentence, seen on lines 524-526, on page 22:

“Relatedly, we did not collect data on duration of recovery in rec-AN, which may be of interest to explore in future research exploring the association between rec-AN and inflammatory markers.”

  1. In contrast of the previous reports, including their own, authors did not find elevated levels of pro-inflammatory cytokines in AN or recovered group.  Though authors have anticipated the role of subjects with autoimmune conditions in this regard, they could not reflect much on this part of the results. It would be great to have a fourth group of patients with AN+autoimmune condition to account for the unanswered questions.

Or authors can present a table showing inclusion and exclusion criteria of the previous studies that report the opposite trend for these pro-inflammatory cytokines to give readers a better insight in this direction.

We have included a sentence that clarifies that only one previous publication in the area has accounted for autoimmune conditions; patients with autoimmune conditions were excluded in a previous publication by Dalton et al., 2018. Also, we agree that including a group of people with AN and an autoimmune condition would be an interesting comparison. However, unfortunately we do not have this data available. We have however included this as a potential direction for future research. These can be seen on lines 491-495, on page 21: “Additionally, as AN and autoimmune conditions have overlapping epidemiology [26], our stringent exclusion criteria of the presence of an autoimmune condition may have contributed to the inconsistency of our findings compared to previous studies, as the majority have not considered autoimmune conditions in their study design or analyses (with the exception of one; [10]). Future research should consider including an additional group of participants with AN and a comorbid autoimmune condition to explore this and the role of inflammatory markers in the relationship between autoimmune conditions and eating disorders”

  1. ANCOVA has to be expanded at the first point of mention.

Thank you for pointing this out, it has been amended in the manuscript.

  1. Typo- page 6, line 279... It should be 'different' in place of 'difference'. 

Thank you for pointing this out, it has been amended in the manuscript.

Reviewer 2 Report

The authors assayed the concentrations of 37 pro-inflammatory cytokines and chemokines in serum to identify inflammatory markers that characterize the disease in patients with anorexia nervosa (AN), recovered patients (rec-AN), and healthy controls. After taking into account variables such as body mass index, degree of depressive symptoms, age, ethnicity, smoking status, and psychotropic medication status, the authors found that blood MIP-1b levels were lower in patients with AN and rec-AN than in healthy controls and were trait markers of the disease. They also found that blood levels of IL-7 and IL-12/IL-23p40 were lower in patients with AN than in healthy subjects, and proposed that these cytokines are state markers.

While much of the other publications show that inflammatory cytokines are increased in patients with anorexia nervosa, the results of this paper showed the opposite. The authors quantified as many as 37 different cytokine levels in the blood, but did not analyze other factors such as electrolyte balance or the ratio of cell types in the blood, so the data and discussion are somewhat lacking and not very convincing.

In patients who repeatedly overeat and vomit, the disruption of electrolyte balance by vomiting may cause disturbance of organs and intestinal microflora. LPS derived from intestinal bacteria may stimulate immune cells to produce pro-inflammatory cytokines and chemokines. If aspiration occurs during vomiting, pneumonia may occur. In this paper, the suppression of inflammatory cytokine production in patients with anorexia nervosa was analyzed in terms of depressive tendencies and medication use as related factors, but a pathological description of the condition of the patient group seems necessary.

However, this paper is worth publishing for the following reasons.

・The main new finding of this paper is that blood levels of IL-7, IL-12/IL-23p40 are lower in patients with anorexia nervosa than in healthy subjects, and these may be potential state markers. Low TNF-a levels in convalescent AN patients were also suggested to be a potential state-marker. The chemokine MIP-1b was suggested to be a potential marker of disease characteristics in patients with anorexia nervosa, providing a candidate where no inflammatory marker has been characterized in this disease.

・It is considered valuable in that the subjects analyzed were human, and more than 100 subjects were analyzed in conditions that varied from patient to patient, such as ethnicity, psychopathology, depressive symptoms, and PTSD.

Specific comments

・Line 28, IL-12/Il-23p40 should be changed to IL-23p40.

・Line 65, 147, 150 and 152, BMI 18.5 kg/m2 should be corrected to m2(superscript).

・Line 201, ....the essay should be corrected to assay.

Author Response

While much of the other publications show that inflammatory cytokines are increased in patients with anorexia nervosa, the results of this paper showed the opposite. The authors quantified as many as 37 different cytokine levels in the blood, but did not analyze other factors such as electrolyte balance or the ratio of cell types in the blood, so the data and discussion are somewhat lacking and not very convincing.

In patients who repeatedly overeat and vomit, the disruption of electrolyte balance by vomiting may cause disturbance of organs and intestinal microflora. LPS derived from intestinal bacteria may stimulate immune cells to produce pro-inflammatory cytokines and chemokines. If aspiration occurs during vomiting, pneumonia may occur. In this paper, the suppression of inflammatory cytokine production in patients with anorexia nervosa was analyzed in terms of depressive tendencies and medication use as related factors, but a pathological description of the condition of the patient group seems necessary.

We acknowledge that we have not measured other factors in the blood, such as leukocyte count, or electrolytes. We have now acknowledged this as a limitation, seen on lines 530-536 on page 22: “In this regard, we would additionally like to emphasise that we did not investigate important cellular parameters of the immune system such as electrolytes and the leukocyte count [67], and more specifically the concentration of Th1, Th2, Th17 and Tregs [68], which play key roles in the immune defence against infectious diseases. Indeed, it has been found previously that individuals with AN have lowered levels of monocytes, lymphocytes and neutrophils [69].”.

We agree that binge/purge behaviours are likely associated with an immune response. In our study, only 9/56 patients with AN were of the AN-BP subtype (see Table 1 in the manuscript). We also excluded participants who reported a recent infection. We now realise that this exclusion was not apparent in the original manuscript, so this has been added in on page 4, line 155 to: “Exclusion criteria for all participants were as follows: male sex; under the age of 18; having any acute or chronic inflammatory or autoimmune condition; self-report of recent infection; current pregnancy; excessive alcohol consumption (>3 units/day, 5 days a week) and/or cigarette consumption (>15 cigarettes/day).”Taken together, we feel that these would have minimal effects on our findings. However, we agree it is important to highlight this as a potential limitation of our study, seen on lines 514-517 on page 22: “Similarly, we included AN participants with both AN-restricting and AN binge/eating purging type. “While eating disorder behaviours associated with AN-BP may be associated with alterations in pro-inflammatory markers via effects on the gut microbiome [66], only nine (out of 56) AN participants in our sample were diagnosed with AN-BP”.

However, this paper is worth publishing for the following reasons.

・The main new finding of this paper is that blood levels of IL-7, IL-12/IL-23p40 are lower in patients with anorexia nervosa than in healthy subjects, and these may be potential state markers. Low TNF-a levels in convalescent AN patients were also suggested to be a potential state-marker. The chemokine MIP-1b was suggested to be a potential marker of disease characteristics in patients with anorexia nervosa, providing a candidate where no inflammatory marker has been characterized in this disease.

・It is considered valuable in that the subjects analyzed were human, and more than 100 subjects were analyzed in conditions that varied from patient to patient, such as ethnicity, psychopathology, depressive symptoms, and PTSD.

Specific comments

・Line 28, IL-12/Il-23p40 should be changed to IL-23p40.

Thank you for pointing this out, it has been amended in the manuscript.

・Line 65, 147, 150 and 152, BMI 18.5 kg/m2 should be corrected to m2(superscript).

Thank you for pointing this out, it has been amended in the manuscript.

・Line 201, ....the essay should be corrected to assay.

Thank you for pointing this out, it has been amended in the manuscript.